# The Effect of Rotavirus Vaccine on Socioeconomic Differentials of Paediatric Care Due to Gastroenteritis in Swedish Infants

**DOI:** 10.3390/ijerph16071095

**Published:** 2019-03-27

**Authors:** Lina Schollin Ask, Can Liu, Karl Gauffin, Anders Hjern

**Affiliations:** 1Sachs’ Children and Youth Hospital, South General Hospital, 118 83 Stockholm, Sweden; lina.schollin-ask@sll.se; 2Department of Medicine, Clinical Epidemiology Unit, Karolinska Institutet, 171 76 Stockholm, Sweden; can.liu@ki.se; 3CHESS, Centre for Health Equity Studies, Stockholm University and Karolinska Institutet, 106 91 Stockholm, Sweden; karl.gauffin@su.se

**Keywords:** inequity, children, gastroenteritis, vaccine, rotavirus, intervention, social disparities

## Abstract

*Background*: Previous Swedish studies have shown a social gradient on paediatric care for viral gastroenteritis. *Aim*: To study the effect of a free rotavirus vaccine programme on hospital care for viral gastroenteritis. *Method*: A register-based national cohort study of paediatric in- and outpatient care for viral gastroenteritis in children <2 years old in two Swedish counties in 2014–2017, with the rest of the country as comparison. Adjusted hazard ratios were estimated by the differences-in-differences (DiD) estimator in Cox regression in the entire cohort and by social indicators. *Results*: Reductions of 37% and 24% for inpatient care, and 11 % and 21% for outpatient care for viral gastroenteritis were found in the Stockholm and Jönköping counties, respectively, after adjusting for time trends and social indicators. For inpatient care, the change was similar over social groups in both counties. In the larger county of Stockholm, smaller reductions in outpatient care were detected for children in socially disadvantaged families. *Conclusions*: A free rotavirus vaccination programme moderately reduced paediatric care for viral gastroenteritis. There were indications of an increase in socioeconomic differences in paediatric outpatient care for viral gastroenteritis, but further studies are needed to confirm this result in a broader health care perspective.

## 1. Introduction

Children from disadvantaged socioeconomic groups have a higher risk of poor health outcomes. This link is well documented, and has been shown in the context of paediatric care for viral gastrointestinal infections [1,2]. Improving child health outcomes and closing the existing social gap in children’s health are identified as the most important factors to combat health inequity, according to the WHO Commission on Social Determinants of Health [3]. Immunisation programmes may be one of many suitable interventions to improve child health outcomes, but evidence on the impact of immunisation programmes is scarce in the literature. 

Globally, viral gastroenteritis is one of the biggest health problems among young children, leading to high mortality in low-income countries [4], and high morbidity with hospital care in high-income countries such as Sweden [5,6,7]. Rotavirus causes the most severe cases of viral gastroenteritis that sometimes lead to hospital admissions [8]. A vaccine against rotavirus was developed in the early 2000s and had been introduced in over 90 countries worldwide by 2017 [9]. Although this vaccine has been available in Sweden since 2006, it has not reached widespread use because it has not been part of the national immunisation programme and requires the cost of the vaccine to be covered exclusively by parental out-of-pocket payments. In 2014, the Swedish county councils in Stockholm and Jönköping included the vaccine in tax-funded regional immunisation programmes, thereby making the rotavirus vaccine available free of charge, with vaccination starting at the age of six to twelve weeks. In Stockholm, a rotavirus vaccine coverage of 85% was reached in 2016 and 90% in 2017 [10,11]. In Jönköping, a coverage of 81% was reached in 2017 [12]. 

In Sweden, a social gradient of both in- and outpatient hospital care due to viral gastroenteritis in infants was described previously, showing a higher hospital care utilisation in groups of lower socioeconomic position (SEP) compared with groups of higher SEP [2]. Large reductions of hospital care for rotavirus-specific gastroenteritis infections as well as all-cause gastroenteritis infections have been shown in many countries after the introduction of a rotavirus vaccine [13]. There are also studies of the vaccine’s effect on the socioeconomic distribution of viral gastroenteritis in the child population within high-income contexts, but the few existing studies on this subject have presented diverging results [14,15,16]. In Sweden, primary care for children is provided by general practitioners, but in some counties publicly-funded specialised paediatric outpatient care is also available without referral, particularly in some university cities like Stockholm. This study aimed to investigate the extent to which introducing the rotavirus vaccine free of charge, as part of a regional immunisation programme, affects incidence of and socioeconomic inequalities in paediatric in- and outpatient care for viral gastroenteritis in a population of Swedish infants. 

## 2. Methods

### 2.1. Study Design and Study Population

This was a register-based study of a Swedish national cohort of 537,479 children aged 2 months to 2 years old, born alive between March 2011 and December 2015, followed up until 31 March 2017. The study exploited the introduction of the rotavirus vaccine in 2 out of 21 Swedish counties in 2014: Stockholm, the largest metropolitan area in Sweden, and Jönköping, an average-sized county. The children were linked by their unique personal identity numbers, assigned to all Swedish residents [17], to their parents through the Multi-generation Register to further retrieve parental and familial characteristics.

### 2.2. Outcomes

Data on paediatric inpatient and outpatient care was retrieved from the Swedish National Patient Register, a register with a high coverage and a high quality of diagnostic information [18]. The outpatient care information included paediatric hospital care in the emergency department as well as publicly-funded paediatric care outside of the hospital. Viral gastroenteritis was defined as a primary or secondary diagnosis of A08–A09 according to the 10th International Classification of Disease (ICD-10). Gastroenteritis cases were excluded if they had a co-existing diagnosis of infection due to non-viral pathogens (ICD-10 codes A04–A07). These diagnostic criteria were used to define two outcomes: *inpatient* care and *outpatient* care. 

### 2.3. Covariates

The following covariates were retrieved from the Swedish Medical Birth Register: gender of the child, maternal age at the birth of the child (<25 years or >25 years old) and cohabitation status of the parents in early pregnancy (yes or no). Parental country of birth was retrieved from The Register of Total Population. Maternal and paternal level of education and use of social welfare benefits were retrieved from the Longitudinal Integration Database for Health Insurance and Labour Market Studies (LISA), held by Statistics Sweden. Level of maternal and paternal education was presented in three categories: less than 9 years (primary education), between 9 and 12 years (secondary education), and more than 12 years (tertiary education). 

### 2.4. Statistical Analysis 

Hazard ratios (HRs) were estimated by Cox regression, with a follow-up time from 2 months after birth until 2 years of age. Each individual child was monitored until their death, the end of the follow-up period in March 2017, their second birthday, or until a first record of in- or outpatient care for gastroenteritis as defined above occurred. The analysis was adjusted for maternal age at childbirth, sex of the child, preterm birth, birth weight in continuous form, education level of the mother and the father, the use of the social welfare benefit, single parent household, and whether both parents were born in a foreign country. 

We used the difference-in-differences (DiD) estimator to estimate the effect of a vaccination programme by comparing children born in counties that implemented the programme with those that did not [19]. The analysis was performed separately for Stockholm and Jönköping counties, due to their different start dates of rotavirus vaccination: cohorts born in March 2015 or later in Stockholm, and cohorts born in May 2015 or later in Jönköping. The children were divided into groups referred to as either “Born Before Vaccine” or “Born After Vaccine” to compare paediatric care in the periods before and after the vaccine introduction.

We modelled the hazard of hospitalization with the Cox regression model:(1)$$⋋\left(\frac{t|\;treatment,\;period)}{exposure}\right)=\beta_{0}+\beta_{1}\times treatment+\beta_{2}\times period+\beta_{3}\times treatment\times period.$$

The DiD estimator was equivalent to the interaction between “Born After Vaccine” versus “Born Before Vaccine” (reference group), and between being born in a vaccinated county council versus the rest of Sweden (reference group). The DiD estimator is the multiplicative interaction term on the hazard ratio scale Exp(*β*_3_) (Figure 1). An Exp(*β*_3_) > 1 suggests that there is a super multiplicative interaction, which indicates the vaccinated county had a higher than expected incidence rate of the disease in the After Vaccine period. In contrast, a submultiplicative interaction suggests that the vaccinated county had a lower than expected incidence rate of the disease in the After Vaccine period. 

When we further analysed the subgroups based on social factors, we used the same analytical approach to compare the differences between the social groups before and after the vaccine programme was initiated in Stockholm and Jönköping, respectively. The social indicators used in this analysis were maternal and paternal education (primary vs. tertiary and secondary vs. tertiary education), maternal age at the birth of the child (<25 years versus >25 years old), usage of welfare benefits, and whether both parents were foreign born versus having at least one Swedish-born parent. The adjustment for covariates was the same as mentioned above. 

### 2.5. Sensitivity Analysis

The children born right before the start of the vaccine programme in Stockholm and Jönköping counties might have been protected indirectly by herd immunity to a certain degree during the follow-up period, which would lead to an underestimation of the true effect of the vaccination. Therefore, we performed a sensitivity analysis in the Stockholm cohort with a 2 year washing off period before the vaccine programme started. The “Born Before Vaccine” birth cohort was set as 1 March 2009 to 1 March 2012, while the “Born After Vaccine” cohort was born in March 2015 or later in Stockholm, and born in May 2015 or later in Jönköping. The statistical analysis was made for the same age period and in the same way as described above.

### 2.6. Ethical Approval

The study was approved by the ethics committee in the Stockholm region in 2015 (dnr 2015/2113-31/5) with a complementary approval in December 2016 (dnr 2016/2380-32) and April 2017 (dnr 2017/732-32). 

## 3. Results

Table 1 describes the socio-demographic characteristics of the study population. Maternal age > 25 years at child birth, a higher parental education level, single cohabitation status, and two foreign-born parents were more common in Stockholm County than the rest of Sweden. In comparison, receiving social welfare benefits was more common in the rest of Sweden compared to Stockholm County.

### 3.1. The Vaccine’s Effect on Paediatric Care Due to Gastroenteritis

Figure 1 and Figure 2 present an overall continuous decrease of both inpatient (Figure 1) and outpatient (Figure 2) paediatric care due to viral gastroenteritis during 2011–2016, the years in the study when we could include comparable whole years of hospital care. However, the slope was steeper for Stockholm and Jönköping during 2015–2016. 

As Table 2 shows, the introduction of the vaccine in the Stockholm County was associated with a reduction of inpatient paediatric care by HR 0.63 (95% CI 0.56–0.70) and for outpatient care by HR 0.89 (95% CI 0.85–0.93) compared to the rest of Sweden, excluding Jönköping. The pattern was similar in Jönköping county with reductions in paediatric care by HR 0.76 (95% CI 0.58–0.99) for inpatient care and HR 0.79 (95% CI 0.69–0.90) for outpatient care in comparison with the rest of Sweden, excluding Stockholm. Both outcomes were adjusted for covariates. 

### 3.2. The Vaccine’s Effect on the Social Gradient of Paediatric Care Due to Gastroenteritis

Table 3 describes the change in paediatric care in Stockholm County after the introduction of the vaccine, stratified by social indicators. For outpatient care, the social gradient increased with estimates ranging from HR 1.15 to 1.39. For inpatient care, the change was similar over social groups. For Jönköping County, only low maternal age was associated with an increase in outpatient care, HR 1.45 (1.06–1.99), see Appendix A.

### 3.3. Sensitivity Analysis

In the sensitivity analyses of Stockholm County, presented in the Appendix A, the change in paediatric care utilisation after the introduction of the rotavirus vaccine was compared with an older cohort born from 2009 to 2012, to avoid the influence of herd immunity. For inpatient care in Stockholm County the HR was 0.64 (95% CI 0.57–0.71) and for outpatient care the HR was 0.92 (95% CI 0.88–0.96) in the sensitivity analysis, consistent with the main analysis. 

## 4. Discussion

In this study, we investigated the extent to which offering a free rotavirus vaccine as part of the regional immunisation programmes in two Swedish counties would affect incidence and socioeconomic inequality in paediatric care for gastroenteritis in an infant cohort. In Stockholm County, we found a moderate decrease of both inpatient care (37%) and outpatient care (11%) due to gastroenteritis in children under the age of two years, after adjusting for general trends in other Swedish counties. The pattern was similar in Jönköping county, but with less-precise estimates because of the smaller population there. Somewhat in contrast to the expectations, we found that socioeconomic inequalities in outpatient care widened after introducing the free vaccine programme. The social gradient in inpatient care was unaffected. 

In other high-income countries, the effect of the rotavirus vaccine on the reduction of inpatient care due to all-cause gastroenteritis has been estimated to be between 42% and 75% [13]. In comparison, our results presented a somewhat smaller reduction of 37%. Our estimate for decrease in outpatient care, 11%, was also lower than a recent study from the UK [14] that found a reduction of 23%. The lower estimates found in our study may have been the result of the adjustment made for the general reduction in paediatric care over time in Sweden—an adjustment that has rarely been made in previous studies. However, our results may also be an underestimation of the total vaccine effect, due to the short follow-up period of only three winter seasons with limited herd immunity and a reported lower vaccine uptake in Stockholm County Council in the first cohorts offered the vaccine compared to later cohorts [10,11].

In Sweden, a social gap in the utilisation of inpatient care for gastroenteritis in small children before the introduction of the rotavirus vaccine has previously been described [2]. In terms of inpatient care, the rotavirus vaccine programme did not have any effect on the social gradient. For outpatient care however, the socioeconomic inequalities in Stockholm increased slightly.

We can only speculate at what caused the increase in social gap in paediatric outpatient care for gastroenteritis after introducing the free rotavirus vaccine. Previous studies of rotavirus vaccine in a social context have presented diverging results. Hungerford and colleagues found a greater rotavirus vaccine effect in socially deprived communities when investigating the need for hospital care due to gastroenteritis in the United Kingdom [14]. In contrast, studies from Israel and Canada have presented a lower vaccine effect of rotavirus vaccine on hospital admissions in neighbourhoods with lower socioeconomic position [15,16].

In contrast to these previous studies, we used multiple family SEP indicators in this study and not only a single neighbourhood variable. Therefore, this study could arguably have the potential to yield more precise estimates of the social gradient. Neither of the previous studies reported paediatric outpatient care by social indicators, thus making this study an important addition to the literature in this aspect. 

In Sweden, factors such as a lower vaccine coverage in less socioeconomically privileged groups might explain the lower reduction of inpatient care. Due to the short follow-up period, this could also be an example of health inequalities caused by socioeconomic differences in the diffusion of innovations. For outpatient care, the simultaneous implementation of a choice market reform in Stockholm County [20] that has included paediatric outpatient care since 2014 may have confounded our results by increasing access to paediatric outpatient care, particularly in high-income neighbourhoods. Further studies that also include outpatient visits to general practitioners are needed to estimate the broader effects of the rotavirus vaccine on outpatient care for gastroenteritis. 

The literature suggests that the persisting health inequalities in the Nordic welfare states could partly be explained by the fact that higher socioeconomic groups are usually early adopters of health-promoting technology, behaviours, and healthcare innovations, even when these are offered free of charge [21]. To compensate for this, the County Councils could increase their efforts to specifically target lower socioeconomic groups when introducing new vaccination programmes. This would be important not only to promote health equity, but to contribute to stronger herd immunity in future generations as well. Finally, parental rehydration strategies at home are important to reduce the burden of disease and the need for health care in children with gastroenteritis. The knowledge of such practices may be more common in higher SEP groups, as it is connected to general health literacy. Thus, to bridge the social gap in hospital care due to gastroenteritis, interventions targeting health literacy may be more important than the rotavirus vaccine programmes [22,23]. 

Overall, there was a reduction of gastroenteritis for both paediatric inpatient and outpatient care in all of Sweden in young children during the study period. The reasons for this are unknown, but we speculate that the reduction may be related to organisational changes in hospital care with a reduction of hospital beds, improved treatment of gastroenteritis in emergency departments, and/or improved access to telephone advice.

A strength of our study was the population-based design and the reliable Swedish registers that were used [18], which created minimal attrition and thus reliable results also for socially vulnerable populations [24]. Another strength was the possibility to adjust our analyses to the general trend of reduction in our indicators of viral gastroenteritis in Sweden. We included three winter seasons, thereby including potential seasonal variations of the rotavirus between different years.

A major limitation with our study was the absence of the individual vaccine status in the analysis of social differences of health care utilisation. Further studies are needed to evaluate vaccine effects from a socioeconomic perspective, taking vaccine uptake into account. The population in Jönköping County was too small to allow for meaningful analyses of the effect of the rotavirus programme by socioeconomic indicators. Thus, further studies in larger populations in medium-sized Swedish counties are needed to evaluate this aspect of the vaccination programme in such contexts. 

The process to include the rotavirus vaccine in the Swedish national immunisation programme has been initiated. Our results indicate that such a programme would considerably reduce the burden of disease and health care utilisation for viral gastroenteritis in small children. To decrease the social gradient for health care utilisation for viral gastroenteritis, however, other types of interventions are probably also needed. 

## 5. Conclusions

Overall, the reductions in the utilisation of paediatric care due to viral gastroenteritis in small children due to the rotavirus vaccine in this Swedish setting were moderate compared with previous studies in high-income settings. The socioeconomic differences in the utilisation of paediatric care for viral gastroenteritis increased for outpatient care in the large Stockholm County where such changes could most readily be detected, but not for the most severe cases of gastroenteritis needing inpatient care. Further studies are needed to estimate the effects of the rotavirus vaccine on outpatient care, where primary as well as secondary outpatient care is included.

## Figures and Tables

**Figure 1 ijerph-16-01095-f001:**
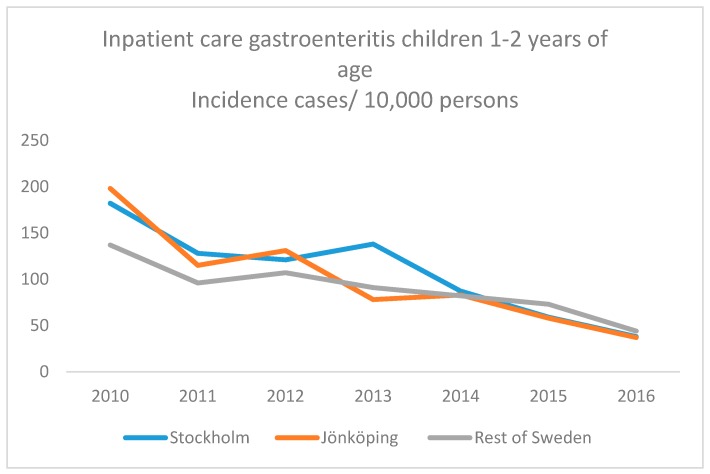
Incidence (incidence cases/10,000 persons) of paediatric inpatient care for viral gastroenteritis in Swedish children aged 12–24 months during 2011–2016.

**Figure 2 ijerph-16-01095-f002:**
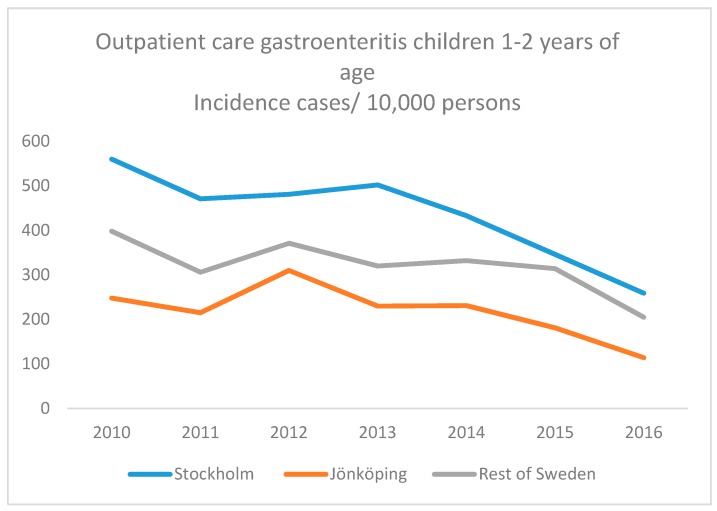
Incidence (incidence cases/10,000 persons) of paediatric outpatient care of viral gastroenteritis in Swedish children aged 12–24 months during 2011–2016.

**Table 1 ijerph-16-01095-t001:** Sociodemographic characteristics of the study population.

Variables	Stockholm County Council		Jönköping County Council		All other Swedish County Councils	
	Children Born Before Vaccine	Children Born After Vaccine	Children Born Before Vaccine	Children Born After Vaccine	Children Born Before Vaccine	Children Born After Vaccine
	(*N* = 84,863)	(*N* = 52,900)	(*N* = 12,369)	(*N* = 6610)	(*N* = 232,827)	(*N* = 147,910)
	(%)	(%)	(%)	(%)	(%)	(%)
**Outcome**						
Inpatient care viral gastroenteritis	2.0	0.8	1.7	0.9	1.5	1.1
Outpatient care viral gastroenteritis	9.1	6.5	5.7	4.0	5.8	5.0
Gender						
Boy	51.5	51.8	51.9	50.5	51.3	51.6
Girl	48.5	48.2	48.1	49.5	48.8	48.4
**Child characteristics**						
*Birth year*						
2011	27.6	0	26.7	0	27.9	0
2012	33.2	0	30.9	0	33.5	0
2013	33.9	0	31.2	0	33.4	0
2014	5.3	46.2	11.2	39.1	5.3	45.6
2015	0	53.8	0	60.9	0	54.4
*Preterm*						
Yes	5.3	5.3	5.7	5.1	5.7	5.6
**Family characteristics**						
*Maternal age at birth of child*						
<25 years	7.8	7.0	14.1	13.5	14.1	12.8
25 or more	92.2	93.0	85.9	86.5	85.9	87.2
*Mother´s highest education*						
Primary	11.5	10.7	12.8	12.9	13.7	13.9
Secondary	7.5	7.2	7.1	7.2	7.6	7.6
Tertiary	81.0	82.1	80.0	79.9	78.8	79.0
*Father´s highest education*						
Primary	16.0	16.0	15.3	16.5	15.9	16.7
Secondary	11.0	9.8	14.6	11.2	13.4	11.5
Tertiary	73.1	74.2	70.1	72.3	70.7	71.8
*Use of social welfare benefit*						
Yes	3.9	3.1	8.7	8.3	9.0	8.6
*Cohabitation of parents during pregnancy*						
No	12.1	12.8	7.1	7.2	9.9	10.2
*Parental country of birth*						
Both parents foreign born	21.8	23.6	17.8	20.4	17.1	19.7
At least one Swedish-born parent	78.2	76.4	82.3	79.6	82.9	80.3

**Table 2 ijerph-16-01095-t002:** Adjusted hazard ratios (HRs) of paediatric inpatient and outpatient care of viral gastroenteritis. Adjusted for parental education, maternal age, social welfare, having two foreign-born parents, and single-parent household status.

Outcome variables of Paediatric Care After vs. *Before* Vaccine start: (*reference in italic*)	Difference-in-Differences Estimate of HR (95% CI)
Inpatient care for gastroenteritis	
Stockholm vs. *Rest of Sweden (Jönköping excluded)*	**0.63** (0.56–0.70)
Jönköping vs. *Rest of Sweden (Stockholm excluded)*	**0.76** (0.58–0.99)
Outpatient care for gastroenteritis	
Stockholm vs. *Rest of Sweden (Jönköping excluded)*	**0.89** (0.85–0.93)
Jönköping vs. *Rest of Sweden (Stockholm excluded)*	**0.79** (0.69–0.90)

Bold text indicates statistical significant associations at the *p* < 0.05 level.

**Table 3 ijerph-16-01095-t003:** Cox regression of the socioeconomic covariates related to paediatric inpatient and outpatient care of viral gastroenteritis in Stockholm County Council.

**Socioeconomic Covariates:**	**Stockholm County Council Difference-in-Differences Estimate of HR (95% CI)**	
**After vs. *Before* Vaccine Start (*Reference in Italic*)**	**Inpatient Care for Gastroenteritis**	**Outpatient Care for Gastroenteritis**
Maternal primary education vs. *Tertiary education*	0.99 (0.74–1.31)	**1.25** (1.12–1.39)
Maternal secondary education vs. *Tertiary education*	1.15 (0.84–1.57)	**1.15** (1.02–1.31)
Paternal primary education vs. *Tertiary education*	0.99 (0.77–1.25)	**1.18** (1.07–1.29)
Paternal secondary education vs. *Tertiary education*	0.81 (0.59–1.12)	**1.14** (1.01–1.28)
Maternal age < 25 at birth vs. *Other*	0.88 (0.62–1.24)	**1.30** (1.15–1.46)
Use social welfare benefit vs. *Other*	0.90 (0.58–1.39)	**1.39** (1.17–1.64)
Both parents foreign born vs. *At least one Swedish-born parent*	0.96 (0.77–1.20)	**1.15** (1.06–1.25)
Single-parent household vs. *Cohabitation*	0.98 (0.76–1.27)	1.10 (1.00–1.23)

Bold text indicates statistical significant associations on the *p* < 0.05 level.

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
