# Peer review of "The Effect of Rotavirus Vaccine on Socioeconomic Differentials of Paediatric Care Due to Gastroenteritis in Swedish Infants"

_ijerph, 2019, doi:10.3390/ijerph16071095_

Round 1

Reviewer 1 Report

The paper presents a study on the effect of Rotavirus mass vaccination on the incidence of outpatient and inpatient hospital access for viral gastroenteritis and the correlation with socioeconomic aspects.

This is an interesting and innovative approach to evaluate the effectiveness of a public health measure in different socioeconomic situation.

There are, however, some points that are not clear and I think some changes could help not Swedish readers.

Some specific comments

Introduction, page 1, lines 40-42.  I would specify which coverage was reached. In the discussion, page 7, lines 183-185, a lower vaccine uptake in the first cohorts is described, but the cited documents are in Swedish and not so easy to find and read. If I understood correctly, the coverage was very high (between 70-78% and 93-96% in 2015-2016) and therefore a herd immunity effect has to be expected.

Introduction, page 1, lines 47-49. The sentence is not clear; does it mean: gastroenteritis general contribution?

Methods, page 2, lines 57-58. Where the children followed up from 2 to 6 years or from 2 months of age until 2 years (see page 2, lines 81-82)? If the follow up was up to 2 years of age, the child was censored after the second birthday (page 2, lines 82-84).

Methods, page 2, line 59. The vaccine was introduced in 2014 or 2015 (see page 3, line 91)? In Table 1, 2014 is considered cohort born after vaccine.

Methods, page 3, line 101. Figure 1 is not related to hazed ratio scale, but to results, page 5, line 135.

Table 1, page 5, last two lines. I imagine that both parents foreign born were 15-19% in other Swedish county councils and not 80-84%.

Discussion, pages 7-8, lines 173-175. The author gave hypothesis to explain both the social differences in outpatient care and the absence of differences in inpatients care and I do not think this is unexpected, moreover in a situation of decrease of gastroenteritis hospitalization.

Author Response

Dear editor

Thank you for reviewing our manuscript entitled “The rotavirus vaccine effect on socioeconomic differentials of health care utilization due to gastroenteritis in small children in Sweden” ID: ijerph-446834).

After the article was submitted, we realized that the parallel introduction of a choice reform for specialized care in the Stockholm County and the subsequent increase in the available pediatric outpatient care, might have confounded our results with regards to outpatient care. We have added some text to the Limitations in the discussion about this and also changed the conclusions in the abstract to point to the need of a broader analysis of the effects of the vaccine on the use of outpatient care.

Additionally figure 1 needs to be moved to the results-section, in the manuscript prepared by the journal editors it has probably by mistake been placed in the methods section. We have revised this placement in the submitted  manuscript.

We have revised our article in light of the comments made by the reviewers.

Response to the reviewers

#1

Thank you for taking the time to read our manuscript so carefully! We have taken the following action based on your comments, responds in italic:

The paper presents a study on the effect of Rotavirus mass vaccination on the incidence of outpatient and inpatient hospital access for viral gastroenteritis and the correlation with socioeconomic aspects.

This is an interesting and innovative approach to evaluate the effectiveness of a public health measure in different socioeconomic situation.

There are, however, some points that are not clear and I think some changes could help not Swedish readers.

Some specific comments

Introduction, page 1, lines 40-42.  I would specify which coverage was reached. In the discussion, page 7, lines 183-185, a lower vaccine uptake in the first cohorts is described, but the cited documents are in Swedish and not so easy to find and read. If I understood correctly, the coverage was very high (between 70-78% and 93-96% in 2015-2016) and therefore a herd immunity effect has to be expected.

We have added information about vaccine coverage in Stockholm and Jönköping to the introduction part according to your comment and added a reference (12) to state the source of this information.

Introduction, page 1, lines 47-49. The sentence is not clear; does it mean: gastroenteritis general contribution?

We have tried to clarify this:

Large reductions of hospital care for gastroenteritis, specific rotavirus infections as well as all-cause gastroenteritis, have been shown in many countries after the introduction of a rota-virus vaccine

Methods, page 2, lines 57-58. Where the children followed up from 2 to 6 years or from 2 months of age until 2 years (see page 2, lines 81-82)? If the follow up was up to 2 years of age, the child was censored after the second birthday (page 2, lines 82-84).

We have tried to clarify this in the methods part.

This was a register based study in a Swedish national cohort of 537,479 children followed up from two months to two years of age.

Each individual child was censored at death, the end of the follow-up period in March 2017, at their second birthday or until a first record of in- or out-patient care for gastroenteritis as defined above, whichever comes first. national cohort of 537,479 children followed up from 2 months to 2 years of age

Methods, page 2, line 59. The vaccine was introduced in 2014 or 2015 (see page 3, line 91)? In Table 1, 2014 is considered cohort born after vaccine.

The rotavirus vaccine was introduced in 2014 in Stockholm and Jönköping counties.

The study exploited the introduction of rotavirus vaccine in 2 out of 21 Swedish counties in 2014; Stockholm, the largest metropolitan area in Sweden and Jönköping, an average sized county

Methods, page 3, line 101. Figure 1 is not related to hazed ratio scale, but to results, page 5, line 135.

That is correct. The results are measured in hazard ratios according to the used cox regression model but the figures show incidences. We have changed the headings of the two figures accordingly.

Table 1, page 5, last two lines. I imagine that both parents foreign born were 15-19% in other Swedish county councils and not 80-84%.

We have changed the label of the last row in the table to At least one foreign-born parent to prevent misinterpretations

Discussion, pages 7-8, lines 173-175. The author gave hypothesis to explain both the social differences in outpatient care and the absence of differences in inpatients care and I do not think this is unexpected, moreover in a situation of decrease of gastroenteritis hospitalization.

We agree.

Reviewer 2 Report

This paper deals about the socioeconomici impact of health care utilisation implementing a free of charge vaccination program for rotiavirus in two Swedish counties.

The study was well disegned: clear and well defined outcomes and covariates, appropriate statistical analysis.

Results are well supported by tables and figures.

Discussion was well conducted and well referenced: limitations of this study were taken into account.

Minor revisions:

line 40: erase double dot

line 189: erase double dot

line 128, at the end: add "of 25 years or more" after Maternal age

line 146: add "for outpatinet care" after 0.90]

line 146: at the end of the sentence add "Both outcomes were adjusted for covariates"

Overall: check for spaces (i.e., line 191, line 213, line 218)

References: standardize the style and check for punctuation mistakes

Author Response

#2.

Thanks for taking the time to read our manuscript! We have the following response.

This paper deals about the socioeconomici impact of health care utilisation implementing a free of charge vaccination program for rotiavirus in two Swedish counties.

The study was well disegned: clear and well defined outcomes and covariates, appropriate statistical analysis.

Results are well supported by tables and figures.

Discussion was well conducted and well referenced: limitations of this study were taken into account.

Thank you!

Minor revisions:

line 40: erase double dot

completed

line 189: erase double dot

completed

line 128, at the end: add "of 25 years or more" after Maternal age

completed

line 146: add "for outpatinet care" after 0.90]

completed

line 146: at the end of the sentence add "Both outcomes were adjusted for covariates"

completed

Overall: check for spaces (i.e., line 191, line 213, line 218)

completed

References: standardize the style and check for punctuation mistakes

completed